# P53 and mTOR signalling determine fitness selection through cell competition during early mouse embryonic development

Sarah Bowling[1,2,3], Aida Di Gregorio[1], Margarida Sancho[1], Sara Pozzi[4], Marieke Aarts[2,3], Massimo Signore[4], Michael D. Schneider [1], Juan Pedro Martinez-Barbera[4], Jesús Gil[2,3] & Tristan A. Rodríguez [1]

Ensuring the fitness of the pluripotent cells that will contribute to future development is important both for the integrity of the germline and for proper embryogenesis. Consequently, it is becoming increasingly apparent that pluripotent cells can compare their fitness levels and signal the elimination of those cells that are less fit than their neighbours. In mammals the nature of the pathways that communicate fitness remain largely unknown. Here we identify that in the early mouse embryo and upon exit from naive pluripotency, the confrontation of cells with different fitness levels leads to an inhibition of mTOR signalling in the less fit cell type, causing its elimination. We show that during this process, p53 acts upstream of mTOR and is required to repress its activity. Finally, we demonstrate that during normal develop- ment around 35% of cells are eliminated by this pathway, highlighting the importance of this mechanism for embryonic development.

[1] British Heart Foundation Centre for Research Excellence, National Heart and Lung Institute, Imperial Centre for Translational and Experimental Medicine, Imperial College London, Hammersmith Hospital Campus, Du Cane Road, London W12 0NN, UK. [2] Cell Proliferation Group, MRC London Institute of Medical Sciences (LMS), Du Cane Road, London W12 0NN, UK. [3] Cell Proliferation Group, Institute of Clinical Sciences (ICS), Faculty of Medicine, Imperial College London, Du Cane Road, London W12 0NN, UK. [4] Developmental Biology and Cancer Programme, Newlife Birth Defects Research Centre, UCL Great Ormond Street Institute of Child Health, 30 Guilford Street, London WC1N, UK. Correspondence and requests for materials should be addressed to J.G. (email: jesus.gil@imperial.ac.uk) or to T.A.Ríg. (email: tristan.rodriguez@imperial.ac.uk)

From the earliest embryonic divisions until the death of the organism, cells are subjected to a remarkable array of pressures that will compromise their fitness. Cell competition is a quality control mechanism that allows the comparison of fitness levels between cells and results in the elimination of those which are viable but less fit than their neighbours. The process has been primarily studied in *Drosophila*, where it is thought to play beneficial roles in the optimisation of tissue fitness during embryonic development[1], the regulation of organ size[2,3] and the maintenance of adult tissue homoeostasis during ageing[4]. In flies, cell competition can also be exploited by cancer cells to eliminate the surrounding wild-type tissue, both to make space for their expansion and fuel their transformation[5–7].

During the early stages of mammalian embryonic differentiation, a wide range of cellular changes take place, including a dramatic increase in the proliferation rate and a rewiring of the transcriptional, epigenetic, metabolic and signalling networks. The dimension of these changes and the requirement for their timing to be carefully orchestrated[8] provides a significant potential for the emergence of aberrant cells that cannot perform these changes efficiently and need to be removed before specification of the germline. Consequently, the onset of gastrulation in mouse is accompanied by a wave of cell death as the epiblast becomes hypersensitive to death signals[9–11].

Cell competition has been suggested to be one of the quality control mechanisms that ensures fitness selection in the early mouse embryo[12,13]. We have shown that during the onset of differentiation, cell competition eliminates defective or mis-patterned cells specifically when they are surrounded by fitter cells[12]. Similarly, cells with lower levels of *c-Myc* than their neighbours[12–14] or higher levels of p53[15] are also eliminated by cell competition in the mouse embryo. Recently in mouse, *Myc*-induced competition has been suggested as a mechanism to safeguard pluripotency[16], as the low *c-Myc* cells eliminated by cell competition were found to be less pluripotent than their high *c-Myc* counterparts. However, although differences in c-*Myc* and p53 are recognised by the cell competition machinery as differences in fitness levels, we do not know what pathways are activated in the mouse embryo downstream of these triggers specifically in a competitive context.

The mechanistic target of rapamycin (mTOR) pathway integrates a variety of extracellular and intracellular signals and functions to control cell growth and metabolism. The mTOR complex 1 (mTORC1, hereafter referred to as mTOR) drives anabolic metabolism in response to positive growth inputs but activates catabolic pathways during starvation[17]. Here we report that mTOR signalling is a key effector of cell competition in the early mouse embryo, as loss of mTOR signalling is both required and sufficient for the elimination of defective cells in a competitive environment. We also find that the tumour suppressor p53 acts upstream of mTOR during this process, and that elevated p53 expression not only labels defective cells as less fit than their neighbours, but also is required for mTOR repression during cell competition. Together, these observations shed light on the pathways that regulate competitive fitness during early mouse development.

## Results

### mTOR is a readout of competition between pluripotent cells.

Here our aim is to identify the pathways that mediate fitness selection during early mouse embryogenesis, and specifically those that respond to relative fitness levels rather than eliminate cells with defects that directly affect their viability. For this we use two different cell models that carry defects that can emerge during early embryogenesis but are not intrinsically cell-lethal:

mis-patterning[18] and karyotypic abnormalities[19]. The BMP signalling defective (*Bmpr1a*[−/−]) cells are mispatterned when surrounded by other defective cells[20], but are eliminated by cell competition when surrounded by wild-type cells[12]. Similarly, tetraploid (4n) cells survive until very late in development or birth when the whole embryo is tetraploid[21–23], but are eliminated by apoptosis when in chimeras with wild-type cells[12]. When referring to these two cell types, we collectively term them as defective or loser cells. To pursue the molecular mechanisms regulating their elimination when surrounded by wild-type cells we use a cell culture system where GFP-labelled defective embryonic stem cells (ESCs) are either co-cultured with wild-type cells or cultured as two separate homogeneous populations (separate culture). The phenotypes present in the co-culture condition, but not in the separate cultures, are deemed to be specific of cell competition. Cells are cultured in serum-free N2B27 media, as this induces differentiation and triggers cell competition[12]. In these conditions, defective cells grow normally in separate culture but are efficiently eliminated from day 3 of co-culture (Fig. 1a).

To identify pathways involved in defective cell elimination, we used microarray profiling followed by Ingenuity Pathway Analysis (IPA), which identifies molecular interactions in omics data. Analysis of the transcriptome of wild-type and BMP-defective cells in separate and co-culture conditions revealed high activation Z-scores for mTOR and related pathways specifically in the co-culture condition (Supplementary Fig. 1A–C), suggesting that mTOR may regulate cell competition in the early mouse embryo. To test this hypothesis, we compared mTOR signalling in wild-type and defective cells in separate and co-culture conditions. We observed by immunofluorescence and flow cytometry analysis that ribosomal protein S6 (S6) phosphorylation, a key signalling readout of this pathway, was decreased in both BMP-defective and 4n cells specifically when they were co-cultured with wild-type cells (Fig. 1b–h and Supplementary Fig. 1D). This starkly contrasted with the robust levels of mTOR activation that control cells showed in separate and co-culture. Furthermore, when cell competition was inhibited by maintaining cells in the naive pluripotent state[12], we observed that mTOR activation in co-culture was similar in defective and wild-type cells (Supplementary Fig. 1E, F), indicating that the loss of mTOR signalling occurs specifically during competitive interactions. Thus, in two unrelated cell competition models, the confrontation with fitter cells leads defective cells to downregulate mTOR pathway activity.

### Reduced mTOR signalling induces less-fit cell elimination.

mTOR plays many roles in the regulation of growth[17] and is required for early post-implantation mouse development[24,25]. We therefore asked if the decrease in mTOR activity observed in loser cells is the reason for their elimination. We have previously shown that caspase inhibition rescues loser cell elimination during cell competition[12]. We found that preventing apoptosis also led to an accumulation of cells with low pS6 staining (Fig. 2a–c), suggesting these would otherwise be eliminated by cell competition. To test whether inhibition of the mTOR pathway is sufficient to induce cell death, we treated differentiating cells with the mTOR inhibitor rapamycin and assessed cell viability. Within 6 h, rapamycin markedly increased the activation of apoptotic markers and by 24 h it led to a 50% reduction in cell numbers (Fig. 2d–f). The cell counts could be rescued by caspase inhibition, indicating that decreasing mTOR activity during ESC differentiation induces death (Fig. 2d), a finding consistent with the requirement for mTOR signalling for post-implantation embryo survival[24,25]. Our observations were not due to changes in mTORC2 pathway activation, as levels of pAKT[T473], a direct

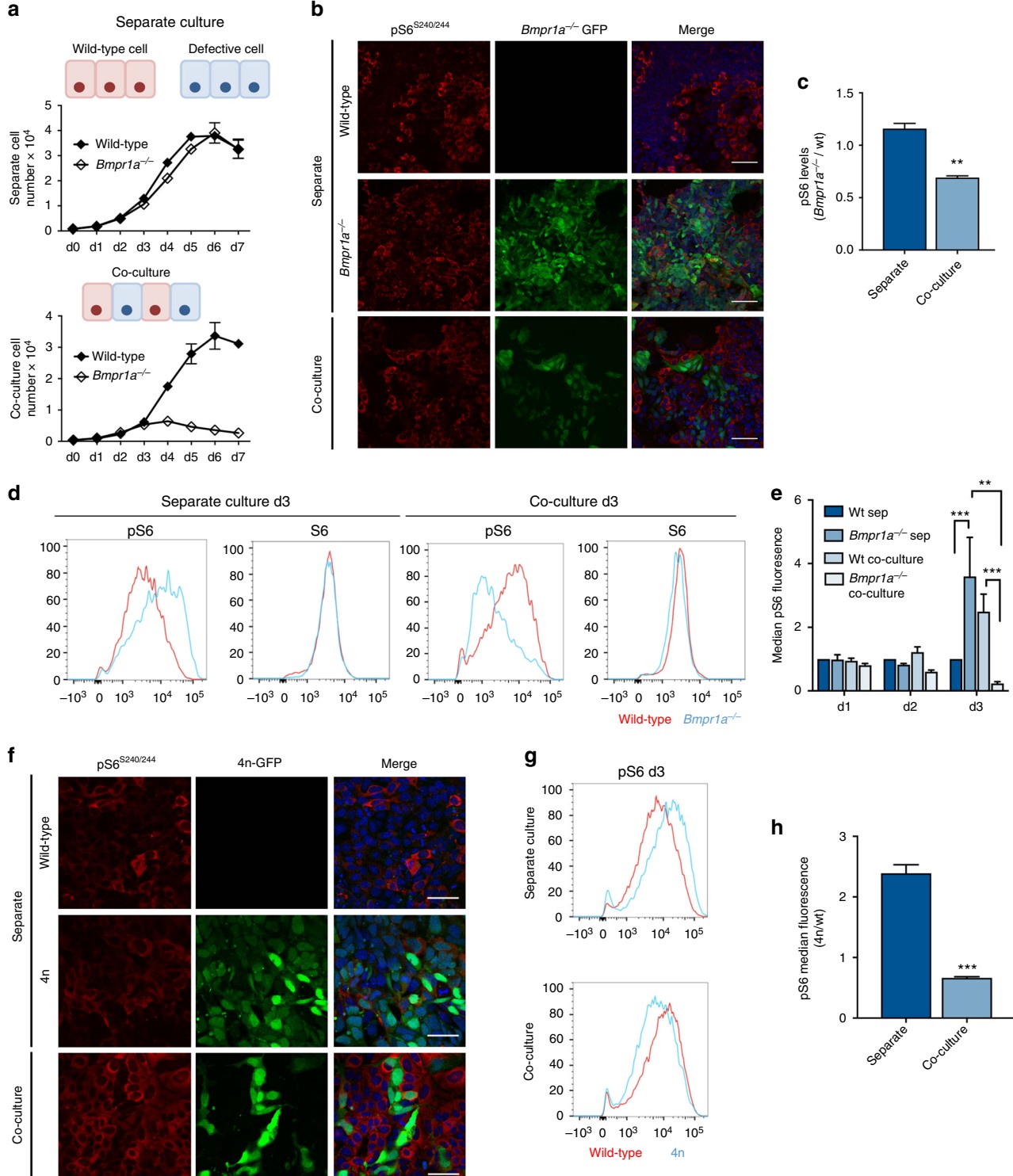

**Fig. 1** Reduced levels of mTOR pathway activity in defective cells during cell competition. **a** Growth curves of wild-type and $Bmpr1a^{-/-}$ cells over 7 days in separate and co-culture. **b** Phospho-S6$^{S240/244}$ levels in wild-type and $Bmpr1a^{-/-}$ cells cultured separately and together were assessed by immunofluorescence analysis and quantified (**c**). Scale bar = 200 μm. **d** Total (S6) and phospho-S6 (pS6) levels in wild-type and $Bmpr1a^{-/-}$ cultured separately and together were assessed by flow cytometry, and the median fluorescence of $Bmpr1a^{-/-}$/wild-type cells was compared in separate and co-culture over the first three days in competition (**e**). **f** Levels of pS6$^{S240/244}$ in wild-type and tetraploid (4n) cells after three days in competition were assessed by immunofluorescence. Scale bar = 50 μm. **g** Levels of pS6$^{S240/244}$ levels in wild-type and 4n cells after three days in competition were assessed by flow cytometry and the median fluorescence of 4n/wild-type cells was compared in separate and co-culture at day three days of competition (**h**). $n > 3$; error bars denote standard error of the mean (SEM). **$p < 0.01$ and ***$p < 0.005$; unpaired, two-tailed $t$-test (**c**, **h**) or ANOVA and Tukeys post-hoc test (**e**)

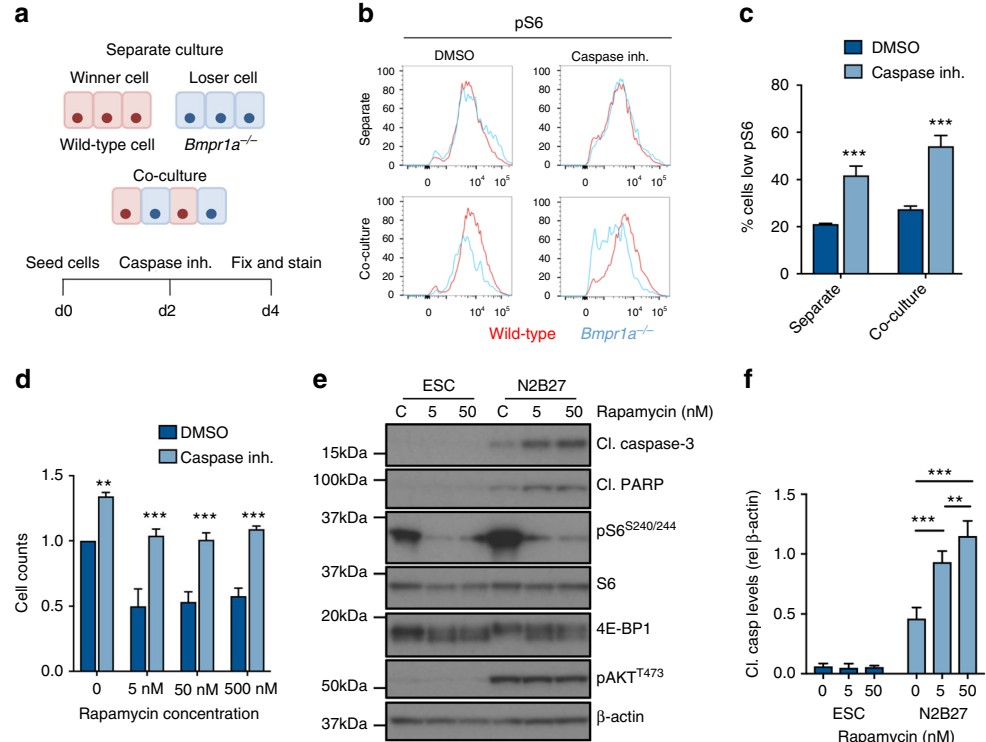

**Fig. 2** Repression of the mTOR pathway triggers cell death in differentiating ESCs. **a** Schematic of experimental setup whereby cells were cultured separately and together for two days in N2B27 before being treated with 100 μM caspase inhibitors for 48 h. Cells were then fixed and levels of pS6$^{S240/244}$ were assessed. **b** Levels of pS6$^{S240/244}$ were assessed by flow cytometry in wild-type and $Bmpr1a^{-/-}$ cells following 48 h treatment with DMSO or caspase inhibitors (100 uM), and **c** median fluorescence of $Bmpr1a^{-/-}$ cells was quantified. **d** Wild-type cells were cultured in N2B27 for 2 days, treated with mTOR inhibitor rapamycin for 24 h and cell count was assessed following treatment with and without caspase inhibitors. **e** Wild-type cells cultured in ESC media, to maintain pluripotency, and N2B27, to initiate differentiation, were treated with rapamycin for 6 h and levels of cell death were assessed by western blot analysis of cl. caspase-3 and cl. PARP. **f** Quantification of cleaved caspase-3 levels relative to β-actin. Error bars denote SEM. *$p < 0.05$, **$p < 0.01$ and *** $p < 0.005$; unpaired, two-tailed $t$-test (**c**, **d**) or ANOVA and Tukeys post-hoc test (**f**)

target of mTORC2, were not altered by rapamycin treatment (Fig. 2e). When naive pluripotent cells were treated with rapamycin no induction of apoptosis was observed (Fig. 2e, f), in agreement with the recent observations that in these cells low mTOR induces a diapause-like state[26]. These results suggest that the loss of mTOR observed in defective cells during competition is directly causing their elimination.

**mTOR is required for less-fit cell elimination**. The above observations suggest that mTOR repression induces defective cell elimination, but we also wanted to address if restoring the activity of this pathway is sufficient to prevent this elimination. For this we targeted *Tsc2* (tuberin), an inhibitor of the mTOR pathway, by CRISPR mutagenesis. $Bmpr1a^{-/-};Tsc2^{-/-}$ clones displayed a loss of TSC2 expression and increased phosphorylation of S6 (Supplementary Fig. 2A). Notably, this increase in mTOR activity induced by TSC2 inhibition completely rescued $Bmpr1a^{-/-}$ apoptotic cell elimination during cell competition (Fig. 3a, b). This was highlighted by the observations that $Bmpr1a^{-/-}$ cell numbers and apoptosis levels in co-culture were restored to those levels seen in these cells in separate culture (Fig. 3a). Importantly, accompanying this rescue of $Bmpr1a^{-/-}$ cell elimination, we observed a change in the behaviour of the wild-type cells the co-culture. We found that when wild-type cells were in a competitive environment with $Bmpr1a^{-/-};Tsc2^{-/-}$ cells, instead of growing exponentially they decreased in cell number (Fig. 3b). This indicates that complete abrogation of TSC2 function makes $Bmpr1a^{-/-}$ cells more competitive than wild-type cells. In

contrast to this, although partial inhibition of *Tsc2* using shRNAs or siRNAs also rescued $Bmpr1a^{-/-}$ cell elimination in co-culture, it did not affect wild-type cell numbers (Supplementary Fig. 2B-F and I), suggesting that the level of Tsc2 inhibition determines the competitive ability of a cell.

TSC2 has been described to regulate exit from pluripotency[27]. To identify whether mTOR hyperactivation through *Tsc2* knockdown was affecting competition through blocking cell differentiation, we analysed expression of pluripotency markers in *Tsc2*-knockdown cells. After three days in N2B27 no differences were observed compared to controls (Supplementary Fig. 2G–H) allowing us to exclude this possibility.

To assess whether mTOR is likewise functionally important in our tetraploid cell competition model we hyper-activated the pathway in 4n cells. We generated two 4n;$Tsc2^{-/-}$ clones by CRISPR that maintained their starting tetraploidy (Supplementary Fig. 3A, B). We observed that deletion of *Tsc2* restored mTOR levels in 4n cells in co-culture (Supplementary Fig. 3C) and completely rescued their apoptotic elimination by wild-type cells (Fig. 3c). Again, accompanying this rescue of 4n cell elimination, we observed that in co-culture, wild-type cells decreased in cell number (Fig. 3d). Analysis of pluripotency markers revealed that although 4n-*Tsc2* null cells present a delay in differentiation, they were still capable of exiting the pluripotent state: at day 3 in N2B27, we observed increased CD31 staining compared with 4n-GFP-targeted controls but by day 4 pluripotency gene expression was similar to controls (Supplementary Fig. 3E, F). Therefore, the rescue we observe is independent of the role of TSC2 in differentiation.

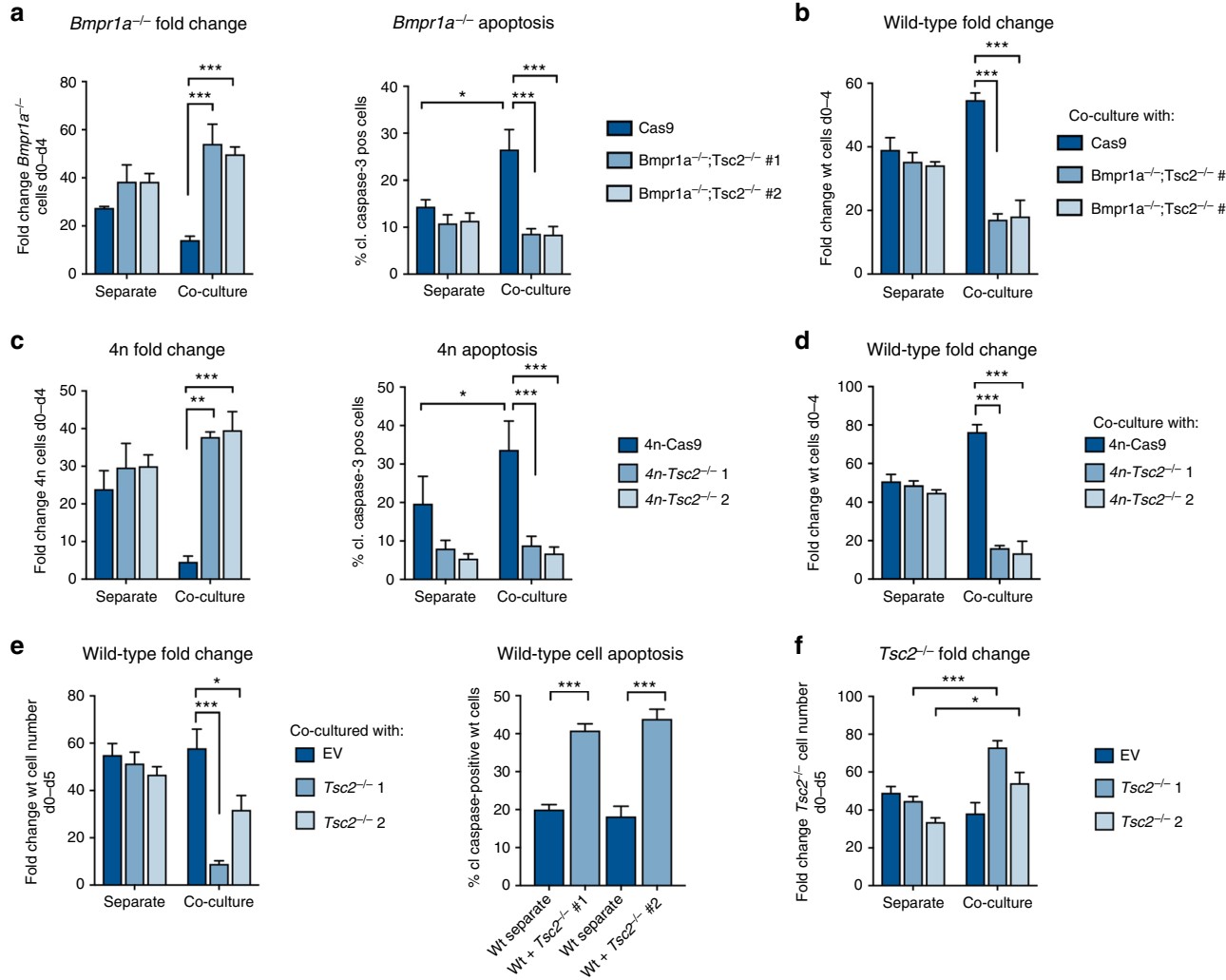

**Fig. 3** Constitutive activation of the mTOR pathway in defective cells rescues their elimination during competition. **a** Fold change in $Bmpr1a^{-/-}$ and $Bmpr1a^{-/-}$; $Tsc2^{-/-}$ (two independent clones) cell numbers between d0-d4 and percent of cleaved caspase-3 positive cells at day 4 cells when cultured alone (separate) or with wild-type cells in N2B27. **b** Fold change in wild-type cell numbers between d0-d4 when cultured separately or co-cultured with $Bmpr1a^{-/-}$ or $Bmpr1a^{-/-}$; $Tsc2^{-/-}$ cells. **c** Fold change in 4n and 4n; $Tsc2^{-/-}$ (two separate independent clones) cell numbers between d0–d4 and percent of cleaved of caspase-3 positive cells at day 4 when cultured separately or with wild-type cells in N2B27. **d** Fold change in wild-type cell numbers between d0–d4 when cultured separately or co-cultured with 4n or 4n; $Tsc2^{-/-}$ cells. **e** Fold change in wild-type cell numbers between d0–d4 and percent of cleaved caspase-3 positive cells at day 4 when cultured separately or co-cultured with Cas9 or $Tsc2^{-/-}$ cells (two independent clones) in N2B27. **f** Fold change in $Tsc2^{-/-}$ cell numbers between d0–d5 when cultured separately or co-cultured with wild-type cells. $n = 3$ for all studies. Error bars denote SEM. $*p < 0.05$, $**p < 0.01$ and $*** p < 0.005$; ANOVA and Tukeys post-hoc test

The observation that 4n-$Tsc2$ null cells can affect the growth of wild-type cells suggests that constitutive activation of mTOR signalling can make cells into super-competitors (i.e. capable of eliminating surrounding wild-type cells). To test this possibility in a diploid background, we deleted $Tsc2$ by CRISPR in wild-type cells and performed competition assays with unaltered wild-type cells. We selected two $Tsc2^{-/-}$ clones with absent levels of TSC2 and hyperactivation of mTOR (Supplementary Fig. 4A). While these $Tsc2^{-/-}$ cells were able to exit naive pluripotency and grew at similar rates to wild-type cells in separate culture (Supplementary Fig. 4B-C), they induced the apoptotic elimination of wild-type cells in co-culture, which showed a doubling in the proportion of cleaved caspase-3 positive cells at day 4 (Fig. 3e). Accompanying the elimination of wild-type cells in co-culture, the $Tsc2^{-/-}$ cells displayed increased growth (Fig. 3f), suggesting they were undergoing compensatory proliferation. These results indicate that mTOR is a key regulator of relative fitness levels, as

not only does the loss of activity of this pathway lead to defective cell elimination, but also increasing mTOR activity is sufficient to trigger competition and transform cells into super-competitors.

**p53 lies upstream of mTOR during defective cell elimination.** We next looked for factors that could regulate mTOR during competition. For this we took a systematic approach and mined our IPA analysis of the transcriptional profiles of wild-type and defective cells for candidates. This study revealed p53, which is known to repress mTOR signalling[28], as the gene with most enriched targets in defective cells (Supplementary Fig. 5A). Furthermore, in *Drosophila*, p53 is required for the metabolic reprogramming of winner cells during cell competition[29] and in MDCK cells, hematopoietic stem cells and mouse embryos an elevation of p53 is sufficient to confer cells with a selective

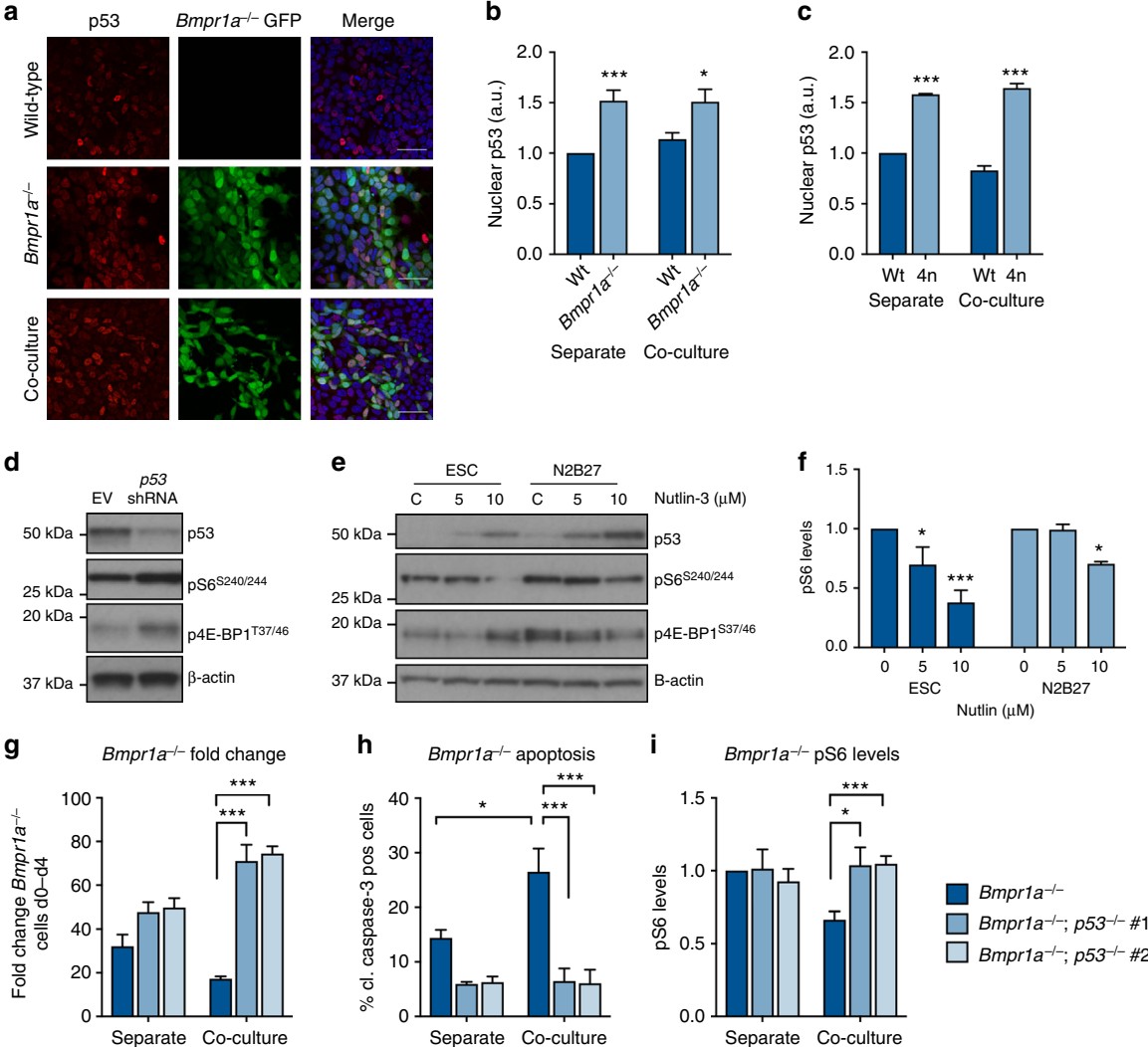

**Fig. 4** p53 is upstream of mTOR pathway repression during ESC differentiation. **a** Total p53 levels in wild-type and *Bmpr1a*$^{-/-}$ cells in separate and co-culture at day 3 in competition were assessed by immunofluorescence and nuclear p53 levels were quantified (**b**). Scale bar = 50 μm. **c** Total p53 levels in wild-type and 4n cells in separate and co-culture were assessed by immunofluorescence and quantified. **d** Wild-type cells were infected with shRNAs targeting *p53* and levels of mTOR pathway activity were assessed by western blot. **e** Wild-type cells were treated with p53 activator Nutlin-3a for 8 h and levels of mTOR pathway activity were assessed by western blot and quantified (**f**). **g** Fold change, apoptosis (**h**), and pS6 levels (**i**) in *Bmpr1a*$^{-/-}$ and *Bmpr1a*$^{-/-}$;*p53*$^{-/-}$ (two independent clones) cultured separately and with wild-type cells (co-culture) from d0–d4. Apoptosis levels of *Bmpr1a*$^{-/-}$;*p53*$^{-/-}$ were compared to *Bmpr1a*$^{-/-}$;Cas9 as previously shown (Fig. 3a) n = 3 for all studies; error bars denote SEM. *p < 0.05, **p < 0.01 and *** p < 0.005; unpaired, two-tailed t-test (**f**) or ANOVA and Tukeys post-hoc test (**b**, **c**, **g**, **h**)

disadvantage[15,30,31]. We observed that expression of p53 and its target p21 were increased in both BMP-defective and 4n cells compared to wild-type cells in both separate and co-culture conditions (Fig. 4a–c, Supplementary Fig. 5B, C). In *Bmpr1a*$^{-/-}$ cells, this increase was not caused by double-strand DNA breaks, as very low levels of gamma-H2AX staining were observed in these cells (Supplementary Fig. 5D). These data suggest that elevated p53 expression is marking defective cells as potential loser cells.

To test if p53 lies upstream of mTOR, we did three things. First, we assessed whether p53 represses mTOR signalling in differentiating ESCs by performing a knockdown of *p53* in wild-type cells using shRNAs and culturing these for three days in N2B27. We observed that inhibition of p53 resulted in an increase in mTOR pathway components pS6 and p4E-BP1 (Fig. 4d). Conversely, we treated wild-type cells with the p53 activator Nutlin-3a and found that increased expression of p53 leads to a repression of mTOR signalling activity (Fig. 4e, f). Neither

treatment of cells with rapamycin nor deletion of *Tsc2* by CRISPR affected p53 levels in *Bmpr1a*$^{-/-}$ cells (Supplementary Fig. 5E-F). Together, these data suggest that p53 activation can inhibit mTOR signalling but that the reverse is not the case.

We then analysed if inhibition of p53 can rescue mTOR levels and defective cell elimination during competition. We initially reduced p53 levels in *Bmpr1a*$^{-/-}$ or 4n cells co-cultured with wild-type cells using siRNAs. In both these cases we observed a partial restoration of mTOR activation levels and a rescue of defective cell growth in co-culture (Supplementary Fig. 6A-F). To test if complete abrogation of *p53* in defective cells led to a more dramatic rescue of their elimination, we generated *Bmpr1a*$^{-/-}$; *p53*$^{-/-}$ cells using CRISPR (Supplementary Fig. 6G). We found that the deletion of *p53* completely rescued both mTOR levels and the elimination of BMP-defective cells in competition with wild-type cells (Fig. 4g–i). These results indicate that p53 lies upstream of mTOR during defective cell elimination by cell competition.

**P53 and mTOR signalling determine cell competition in vivo**. To address the in vivo significance of our observations, we first analysed tetraploid chimeras. Tetraploid embryos are viable until very late in gestation[21–23], but we and others have previously shown that at 6.5 dpc 4n cells are efficiently eliminated in chimeras with 2n cells[12,32]. When we analysed levels of mTOR activity in 4n cells in chimeras at 5.0 dpc, just prior to the stage at which they are eliminated, we found that they display a loss of mTOR activity (Fig. 5a and Supplementary Fig. 7A). This indicated that defective cells also lose mTOR activity prior to their elimination in the embryo. We then tested if hyperactivation of the mTOR pathway is sufficient to rescue defective cell elimination in vivo. Injection of 4n-$Tsc2^{-/-}$ ESCs into diploid embryos gave rise to high contribution chimeras that sustained high mTOR activity (Fig. 5a, b and Supplementary Fig. 7B, C). Significantly, while at 6.5 dpc, 71% of 4n cells in chimeras are apoptotic, only 7% of 4n-$Tsc2^{-/-}$ cells in chimeras were apoptotic ($n = 5$; Fig. 5c)[12]. Furthermore, in 43% of embryos these epiblast cells deformed the epithelial bilayer and overgrew into the proamniotic cavity ($n = 7$; Fig. 5b, d), suggesting that the persistence of 4n cells was disrupting embryonic development. In chimeras with diploid cells, 4n cells rarely survive past gastrulation[32]. However, we observed that 4n-$Tsc2^{-/-}$ cells gave rise to highly chimeric embryos at 7.5 dpc and 9.0 dpc (Fig. 5d, e). In spite of their survival, at 9.0 dpc these embryos suffered a strong developmental delay and presented an abnormal epithelial organisation when compared to their littermates, either because of the negative effects of constitutive mTOR activation on development or because of the negative consequences that the persistence of tetraploid cells has on embryogenesis.

The above results indicated that repression of mTOR signalling leads to defective cell elimination during early mouse development and led us to investigate the frequency of this occurrence during normal development. For this we analysed pS6 levels in embryos cultured either in DMSO or with pan-caspase inhibitors for 16–18 h from 5.5 to 6.25 dpc. This treatment completely abolished TUNEL-positive apoptotic cells and led to a 35% increase in epiblast cell number (Fig. 6a–c). In cells, caspase inhibition results in an accumulation of $Bmpr1a^{-/-}$ cells with low mTOR pathway activity (Fig. 2a–c). Similarly, we observed in vivo that while in control embryos few cells (~3%) showed loss of pS6$^{S240/244}$, in embryos treated with the pan-caspase inhibitor there was a significant increase in the proportion of pS6$^{S240/244}$-negative cells, with about 42% of cells showing low or no staining for this mTOR activation marker (Fig. 6d, e and Supplementary Fig. 8A). Analysis of total S6 levels revealed that those cells that

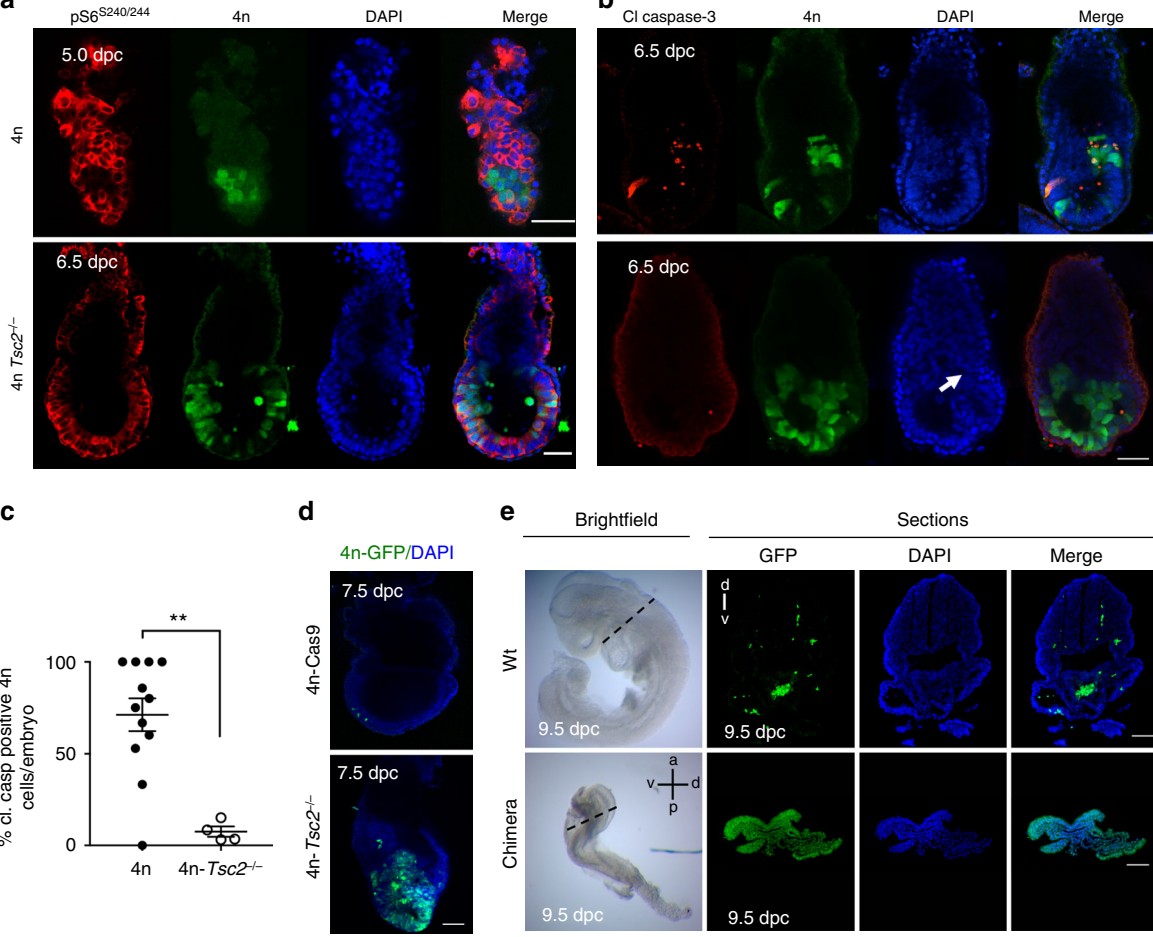

**Fig. 5** mTOR pathway repression is required for competition in vivo. **a** Chimeras generated by injecting 4n and 4n-$Tsc2^{-/-}$ cells into wild-type blastocysts. Levels of pS6 were analysed at 5.0 dpc and 6.5 dpc, respectively. **b** Chimeras generated by injecting 4n ($n = 12$) and 4n-$Tsc2^{-/-}$ ($n = 4$) cells into wild-type blastocysts. The white arrow indicates epiblast overgrowth of 4n-$Tsc2^{-/-}$ cells in these chimeras. Levels of cell death were assessed by cleaved caspase-3 staining at 6.5 dpc and quantified (**c**). **d** Chimeras generated by injecting 4n-Cas9-GFP and 4n-$Tsc2^{-/-}$-GFP cells into wild-type blastocysts. Embryos were analysed for GFP expression to determine contribution of 4n cells. **e** 4n-$Tsc2^{-/-}$ GFP cells were injected into wild-type blastocysts, embryos were collected at 9 dpc, and levels of 4n $Tsc2^{-/-}$ GFP contribution were assessed by whole-mount immunofluorescence or by sectioning. Morphology of chimera is compared to wild-type control embryo. Scale bars are 50 μm for **a**, **b** and 100 μm for **d**, **e**. Error bars denote SEM (**c**). **p < 0.01; t-test

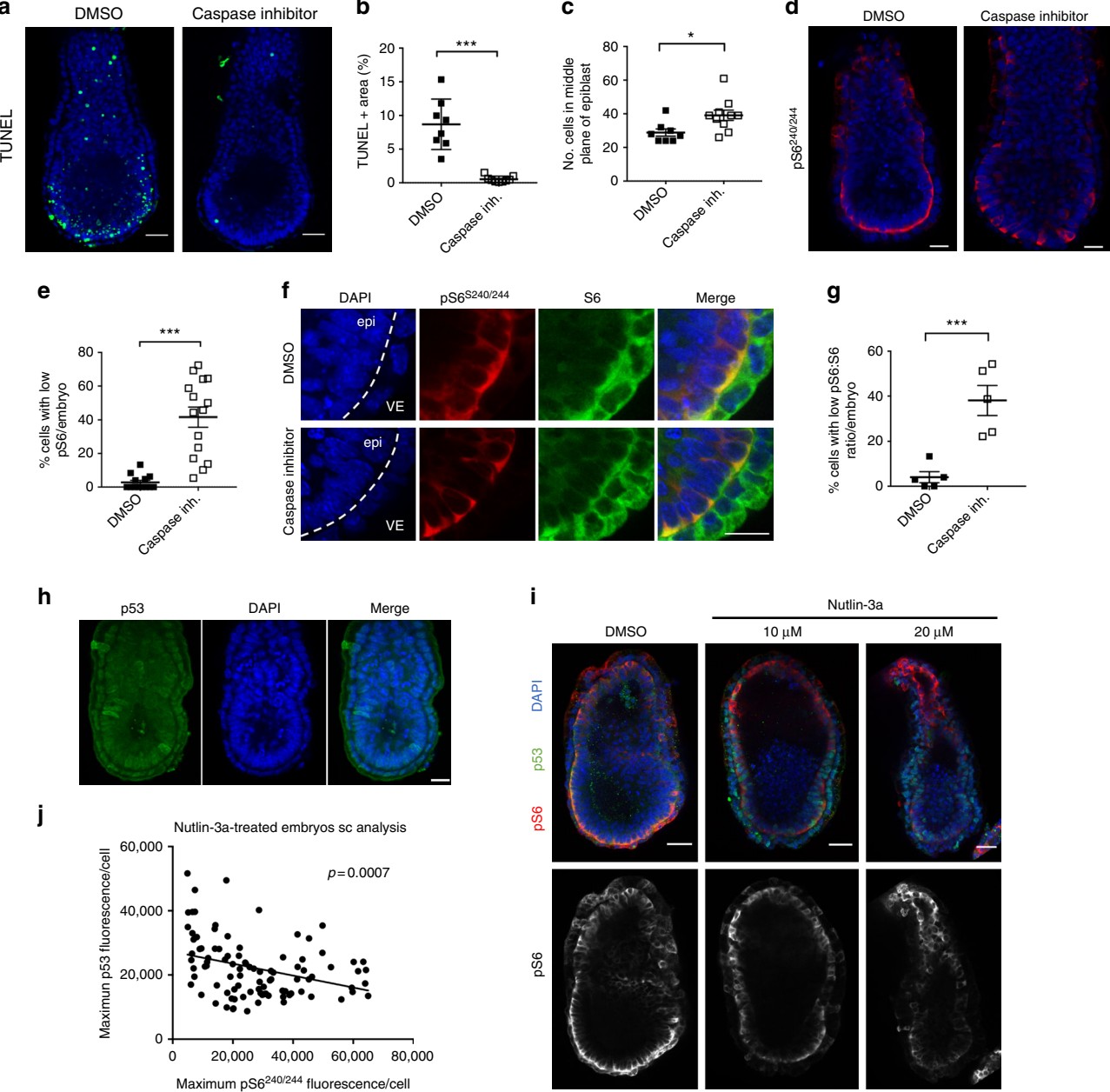

**Fig. 6** Evidence for a role of p53 and mTOR signalling during physiological competition during embryogenesis. **a, b** Wild-type embryos were cultured in DMSO or 200 uM pan-caspase inhibitor for 16–18 h and cell death was assessed by TUNEL staining. **c** Wild-type embryos were cultured in DMSO or caspase inhibitors and number of cells in the medial plane of the epiblast was quantified; caspase inhibition results in a 35% increase in the average number of cells counted in the medial plane of the epiblast (DMSO $n = 8$, average 28.9 cells; caspase inhibitor $n = 10$, average 39.1 cells). **d** Wild-type embryos cultured overnight in DMSO or caspase inhibitors were stained for pS6$^{S240/244}$. **e** In each embryo, the percentage of cells below a 'low pS6 cutoff' in the medial plane of the epiblast was quantified (as shown in Supplementary Fig. 8A): percentage of low pS6 cells increases from 2.8% in DMSO treated embryos to 41.5% in caspase-inhibitor-treated embryos. Data are collated from three litters (DMSO $n = 13$; caspase inhibitor $n = 15$). **f** Magnification of DMSO- or caspase inhibitor-treated embryos stained for pS6 and total S6. The boundary between the visceral endoderm (VE) and epiblast (epi) is indicated with a dashed line. **g** In each embryo, the percentage of cells below a 'low pS6:S6' cutoff (as shown in Supplementary Fig. 8B) was quantified: percentage of low pS6:S6 cells per embryo increases from 4% in DMSO-treated embryos to 38% in caspase-inhibitor-treated embryos. Data are collated from one litter (DMSO $n = 5$; caspase inhibitor $n = 5$). **h** p53 expression in 5.5 dpc embryos detected by immunofluorescence. **i** Embryos were treated with DMSO or Nutlin-3a for 6 h and levels of p53 and pS6 were analysed by immunofluorescence and the correlation between p53 and pS6 levels were quantified at the single-cell level (**j**). Scale bars are 40 μm (**a, i**), 25 μm (**b**) or 20 μm (**f, h**). Error bars denote SD (**b, c**) or SEM (**e, g**). **p < 0.01, ***p < 0.005; unpaired, two-tailed *t*-test

had down-regulated pS6 expression showed robust total levels of S6 protein in the epiblast (Fig. 6f, g and Supplementary Fig. 8B, C), confirming that the loss of mTOR signalling occurring in those cells fated for elimination is not due to a general decrease in protein expression. These results suggest that during early post-implantation development a rapid flux of cell competition eliminates around 35% of cells, and that this elimination likely occurs through repression of mTOR pathway activity.

We next analysed if p53 could also be upstream of mTOR in the embryo. At 5.5 dpc p53 expression is heterogeneous, with a

small proportion of cells showing elevated p53 expression (Fig. 6h). Analysis of the relative p53 and pS6 levels in 5.5 dpc embryos failed to reveal any correlation between the expression of these factors in the unperturbed embryo (Supplementary Fig. 8D). One likely explanation for this is that p53 has a very short half-life and its expression oscillates, with apoptosis being induced only when protein levels are stabilised above a certain threshold[33,34]. Furthermore, how quickly p53 levels increase in the cell also determines if apoptosis is induced or not[35]. For these reasons, to obtain a rapid stabilisation of p53 and analyse how this affects mTOR activity, we have used the Mdm2 inhibitor Nutlin-3a. When 6.5 dpc embryos were cultured in the presence of Nutlin-3a for 6 h we observed a direct correlation at the single-cell level between p53 and pS6 levels (Fig. 6i, j). This suggests that in the embryo, sustained p53 expression is required to repress mTOR. These observations together with the findings that mutation of p53 in 4n cells allows these to give rise to very high chimeras that survive till birth[36], indicate that during early embryogenesis both p53 and mTOR signalling play key roles in defective cell elimination, and strongly suggest that p53 acts upstream of mTOR during this process.

## Discussion

Cell competition is thought to act as a quality control mechanism and has primarily been studied in flies. More recently, fitness selection though cell competition has been shown in the early mammalian embryo at the onset of gastrulation[12,13]. However, the mechanisms marking cells as less fit and those involved in loser cell elimination have remained obscure. Here we start to unravel these mechanisms with three key findings. First that two cell types with very different types of defects (mispatterned and karyotypically abnormal cells) are eliminated via mTOR-dependent cell competition. Second, in both genetic circuits, the tumour suppressor p53 operates upstream of mTOR as an essential regulator of kinase activity. Third, this epistatic relationship between mTOR and p53 was substantiated not only in cultured ESCs, but also in mouse embryos at the onset of gastrulation.

Little is known about mTOR and cell competition. In Drosophila, increasing insulin signalling (an upstream mTOR regulator) by expression of the catalytic subunit of PI3K[3] or mutation of Pten[37] is not sufficient for cells to acquire a winner status and eliminate the surrounding wild-type cells. Similarly Tsc1, Tsc2 or Pten mutations do not rescue the elimination of Minute cells in imaginal wing discs[38]. In contrast to these observations, here we find two things. First, that two cell types with very different types of defects, such as BMP defective and tetraploid are eliminated by mTOR repression. Second, that increasing mTOR activity by Tsc2 mutation makes cells into super-competitors that can eliminate wild-type cells. Together, these results point to the relative levels of mTOR signalling as being a key determinant of the competitive nature of cells during early embryogenesis. Our finding that blocking apoptosis in the embryo results in the accumulation of cells with low mTOR activity from ~1% to ~40%, strengthens the importance of these findings, as they suggest that competition through mTOR is not just a sporadic effect, but occurs at a significant scale during early post-implantation development.

Our studies also reveal that during mouse cell competition mTOR is regulated by p53. P53 is emerging as a key player in cell competition in different systems. In flies p53 is required to sustain the metabolic status of winner cells during competition, as well as for their ability to signal loser cell elimination[29]. In mammals the roles of p53 in cell competition appear to be different. The observations that polarity deficient MDCK cells[30], as well as BMP

defective and tetraploid mouse ESCs, have a non-lethal increase in p53 independently of being in a competitive context suggests that a moderate increase in p53 occurs in response to a wide variety of stresses and acts as a less-fit signature. This signature does not directly affect viability, but instead labels cells as losers when placed in a competitive environment with wild-type cells, as we have found in the mouse embryo with BMPR-defective and tetraploid cells, or as others have observed through mutation of Mdm2/4[15], in haematopoetic stem/progenitor cells subject to DNA damage[31] or in polarity defective MDCK cells[30].

The pathways acting downstream of p53 during cell competition have so far remained elusive, and here by identifying mTOR we gain insight into the molecular mechanisms used by p53 to determine relative cell fitness. Given our data, we envisage two possible scenarios to explain the relationship between p53 and mTOR. First, it is possible that p53 elevation labels defective cells as losers, but when these cells are in a competitive environment a second unidentified factor becomes activated in these defective cells, and this mediates repression of mTOR and their elimination. Alternatively it is possible that during competition, p53 changes in activity to repress mTOR and induce defective cell elimination. Distinguishing between these possibilities will help elucidate if the interaction between p53 and mTOR is direct or not.

Cell competition has been proposed to have roles from the elimination of defective cells in development to promoting the expansion of cancer cells during tumourigenesis[39]. However, the roles of cell competition during mouse development remain unresolved. The recent observation that the low c-Myc cells eliminated by cell competition are of a lower pluripotency status (primed pluripotent), than their high c-Myc counterparts (that are naive pluripotent) has led to the interesting proposal that cell competition is acting to remove cells that exit naive pluripotency prematurely[16]. However, it is important to define precisely at which developmental stage this removal is likely to be occurring. During early mouse embryogenesis, there are two waves of cell death that take place, one at the pre-implantation stage, at about 4.5 dpc[18] and a second during early post-implantation development, at 6.5 dpc[40]. At 4.5 dpc, epiblast cells are in the naive pluripotent state; therefore, it is plausible to argue that the wave of apoptosis occurring at this stage is removing cells that have acquired the primed pluripotent state precociously. In contrast to this, at 5.5 dpc in the early post-implantation embryo, all cells are uniformly primed pluripotent[41,42] and then at 6.5 dpc and 7.5 dpc these cell initiate gastrulation in an asynchronous fashion[8]. This makes it less likely that those cells dying at 6.5 pc are doing so because they have acquired a primed pluripotency status before their neighbours.

The scale of changes occurring during the onset of differentiation, including a doubling of the proliferation rate[43], means there is a significant potential for the emergence of defective cells. Therefore, an alternative possibility is that the cell elimination occurring at 6.5 dpc is because cell competition is acting as a quality control checkpoint to prevent these cells from contributing to the germline, that is specified around this time[44]. One prime example of a cell type that could be eliminated by this mechanism is karyotypically abnormal cells. These are remarkably common in early mammalian development, for example in humans 50–80% of embryos have karyotypically abnormal cells at pre-implantation stages[19]. However, karyotypic abnormalities are not necessarily per se prohibitive for early embryonic development, as even fully tetraploid embryos are viable until very late in gestation[21–23] and they gastrulate at the same time as wild-type embryos[45]. Here we show that cell competition eliminates these tetraploid cells in a mTOR dependent manner at 6.5 dpc, just when the germline is being specified[44]. The observation that

mutation of p53 allows tetraploid cells to give rise to high chimeras that survive till birth[36], strengthens the case for the involvement of p53-mTOR signalling in the elimination tetraploid cells. Therefore, together, these results suggest cell competition may be an important checkpoint for the removal of karyotypically abnormal cells acting before the germline forms to ensure its integrity.

A third possible role for cell competition during early mouse embryogenesis is in the regulation of embryo size. The mouse embryo is remarkably regulative and can compensate for a significant gain or loss of cells. For example, the embryo adjusts to a doubling in cell number at pre-implantation stages[46] or a loss of up to 90% of cells at early post-implantation stages[47] and regulates growth so that by organogenesis embryo size is similar to that of untreated controls. It is possible that by balancing proliferation and cell death, cell competition may coordinate the growth of the rapidly expanding early post-implantation mouse embryo[8], and in this way contribute to this regulative ability of the early mouse embryo. The fact that mTOR is a major metabolic regulator raises the possibility that through p53-mTOR-regulated cell competition, the epiblast cells may be able to integrate nutrient and growth factor inputs and adjust cell numbers to ensure that the rapid acceleration of embryonic proliferation remains within specific parameters. The observation that treating pre-implantation embryos with insulin-like growth factor 1, that is upstream of mTOR, leads to larger embryos by increasing inner cell mass cell numbers[48], is tantalising evidence to suggest that cell competition may indeed play this role.

*Tsc2* null mice die between 9.5 and 12.5 dpc[49,50]. In contrast to this, p53 null mice survive until adult-hood[51], but both p63 and p73 can compensate for its function during early post-implantation development[52]. *c-Myc* is another important regulator of cell competition, and although null mutants die around 10.5 dpc[53,54], *d-Myc* can also compensate for *c-Myc* function during early mouse development[55]. To help uncover the roles of cell competition it will be important to ascertain the levels of cell death in the above mutant strains, and if reduced, to establish the identity of the cells that would normally have been eliminated. In parallel, it will also be interesting to test if mouse mutants for Tsc2, the p53 family or the Myc family can scale properly.

In conclusion, our work has uncovered that p53 regulation of mTOR is an important determinant of cellular fitness during early mammalian embryonic development. The fact that an increase or a decrease in expression of either of these signals induces defective cell elimination or super-competition just highlights this importance. It will be fascinating to uncover how conserved this pathway is in other mammalian tissues, as well as to understand if it plays any role in the fly.

## Methods

**Cell culture**. Wild-type E14 cells were a gift from Prof. A. Smith; $Bmpr1a^{-/-}$ cells and tetraploid cells were obtained from previously described sources[12,20]. No cell lines used here are listed in the ICLAC and NCBI Biosample database of commonly misidentified cell lines.

Cells were cultured at 37 °C and 5% $CO_2$ in GMEM-BHK1 supplemented with 10% FCS, 2 mM glutamine, 1 × MEM non-essential amino acids, 1 mM sodium pyruvate, 0.1 mM β-mercaptoethanol (all Thermo Fisher Scientific) and 100 units/ml leukaemia inhibitory factor (generated and tested in the lab). ESCs were maintained on gelatin-coated flasks and media was changed daily. Cell lines were routinely tested for mycoplasma contamination.

Details of rapamycin (Cayman Chemical), Nutlin-3a (BioVision) and caspase inhibitor (Z-VAD-FMK, R&D Systems) treatments are outlined in figure legends.

**Competition assay**. Cells were seeded onto plates coated with fibronectin (Merck) at a concentration of $2.05 \times 10^4$ cells/cm² either separately or mixed for co-cultures at a 50:50 ratio. The cells were cultured in N2B27 media (Neurobasal media; DMEM F12 media; 0.5 × B27 supplement; 0.5 × N2 supplement; 0.1 mM 2-mercaptoethanol; 2 mM glutamine; all Thermo Fisher Scientific) to allow for differentiation. At the indicated time-points, the cells were counted using ViCell

Counter and Viability Analyser (Beckman Coulter) and proportions of each cell type in co-cultures were determined using LSR II Flow Cytometer (BD Bioscience). Sytox blue (Thermo Fisher Scientific) or propidium iodide (Sigma) was used to stain for dead cells.

**Mice**. All mice were maintained on a 10 hr–14 hr light–dark cycle and were maintained and treated in accordance with the Home Office's Animals (Scientific Procedures) Act 1986. Wild-type mice analysed were on a CD1 out-bred genetic background. Noon of the day of finding a vaginal plug was designated 0.5 dpc/. Embryo dissection was performed M2 media (Sigma). No distinction was made between male and female embryos during the analysis.

**Chimeras**. Embryos were harvested from F1 (C57BL/6 × CBA) or C57BL/6 × F1 crosses. 2.5 dpc embryos were flushed from oviducts in M2 (Sigma) and cultured in BlastAssist (Origio) under embryo-tested mineral oil (Sigma) at 37 °C and 7% $CO_2$ in air. ESCs (3–8) were injected via a laser-generated perforation in the zona pellucida using XYClone (Hamilton Thorne Biosciences).

**Embryo culture**. For caspase inhibitor experiments, embryos were dissected at 5.5 dpc, cultured overnight (18 h) at 37 °C and 5% $CO_2$ in 200 μM pan-caspase inhibitor Z-VAD-FMK (R&D Systems) or the equivalent DMSO volume in poor N2B27 media (Neurobasal media; DMEM F12 media; 0.25 × B27 supplement; 0.25 × N-2 supplement; 0.1 mM 2-mercaptoethanol; 2 mM glutamine; all Thermo Fisher Scientific), and fixed as below. For Nutlin-3a experiments, embryos were dissected at 6.5 dpc, cultured in 10 or 20 μM Nutlin-3a (BioVision) or the equivalent DMSO volume for 6 h in poor N2B27 media, and fixed.

**Immunofluorescence**. Cells were fixed in 4% PFA for 10 min, permeabilised with 0.4% Triton X-100 for 5 min and blocked with 10% BSA (Sigma), 0.1% Triton X-100 for 1 h. Staining with primary antibodies (pS6$^{S240/244}$ #5264, S6 #2317, p53 #2524, cleaved caspase-3 #9664, all Cell Signaling; and λH2AX, Millipore 05-636; all 1:200 dilution) was performed overnight at 4 °C in 1% BSA, 0.1% Triton X-100. After three washes in PBS, secondary antibodies (Alexafluor-488, 546 or 633, Thermo Fisher Scientific; 1:500 dilution) and Hoescht (Thermo Fisher Scientific; 1:1000 dilution) were applied for 1 h. Coverslips were then mounted on glass slides using Vectashield (Vector Labs).

The embryos were fixed using 4% PFA in PBS + 0.01% Triton X-100 and 0.1% Tween. Permeabilisation was performed with 0.5% Triton X-100 in PBS for 20 min, and embryos were blocked using 2% horse serum in PBS + 0.1% Triton X-100 (PBT) for 45 min. Primary antibodies, as above, were incubated with embryos in blocking solution overnight at 4 °C. Following three washes in PBT, the embryos were incubated with secondary antibodies and Hoescht, as above, for 1 h at 4 °C. TUNEL staining was performed using Click-iT Plus TUNEL Assay (Thermo Fisher Scientific). All images were captured on a Zeiss LSM-780 confocal microscope.

Embryo cryosectioning was performed by soaking fixed (4% paraformaldehyde for 6 h at 4 °C) embryos first in 30% sucrose/PBS at 4 °C and then in 7.5% gelatin/15% sucrose/PBS at 37 °C, before being transferred to room temperature to allow the gelatin/sucrose mixture to solidify and blocks to form. Blocks were transferred to 2-methylbutane at -70 °C and stored at −80 °C before sectioning. Sectioning was performed on a cryostat (Leica) using glass blades. GFP was detected using anti-GFP antibody #A-31851 (Thermo Fisher Scientific, 1:50 dilution).

In cells, the levels of cytoplasmic pS6 per field were quantified by measuring average fluorescence intensity in the two pixels surrounding cell nuclei, using Hoescht staining as a mask. Levels of nuclear p53 per field were quantified by measuring average fluorescence intensity in cell nuclei, using Hoescht staining as a mask. This was performed in ImageJ on single sections. In both cases, a minimum of three fields were quantified in each experiment.

For pS6 and p53 quantification in embryos, a line was drawn manually across the nucleus and basal cytoplasm of each cell and the maximum fluorescence intensity across each line was measured. To define low pS6, a cut-off of 20 000 grey values (litter 1; $n = 9$), 12,000 grey values (litter 2; $n = 9$) or 10 000 grey values (litter 3; $n = 10$) was used to define low pS6. We adjusted the cutoff between litters owing to variability in mean fluorescence levels between independent experiments. A ratio cut-off of 0.3 was used to define the low pS6:S6 population. Example graphs showing the cut-off used are presented in Supplementary Fig. 8A-B. For cleaved caspase-3 quantification of tetraploid cells in embryos, cells positive or negative for cleaved caspase-3 were counted manually. In all cases, the analysis was performed in ImageJ on single sections.

**Flow cytometry staining**. Cells were detached from plates using accutase (Sigma) and fixed in 7.4% formaldehyde in N2B27 media for 10 min. Permeabilisation was carried out using ice-cold methanol, and cells were blocked using 1% BSA. The primary antibody (pS6$^{240/244}$ #5264, cleaved caspase-3 #9664, both Cell Signaling; 1:200 dilution) was incubated with cells for 1 h at room temperature. After washing, the secondary antibody was applied (Alexafluor-546/405, Thermo Fisher Scientific; 1:2000 dilution) for 30 min. CD31 (BD Biosciences, MEC13.3; 1:200 dilution) staining was performed on live cells. The antibody was incubated with cells for 45 min at 4 °C and cells were washed in PBS prior to flow cytometry

analysis. All flow cytometry was performed using a BD LSR II flow cytometer and analysed using FlowJo software (both BD Biosciences).

**Western blotting**. Cells lysates were collected using Laemmli buffer (0.05 M Tris-HCl at pH 6.8, 1% SDS, 10% glycerol, 0.1% β-mercaptoethanol), quantified using BCA quantification (Thermo Fisher Scientific) and resolved using Criterion XT pre-cast gels (BioRad) followed by transfer to PVDF membranes. Antibodies used are as follows: pS6$^{S240/244}$ #5264, S6 #2317, p53 #2524, p4E-BP1$^{T37/46}$ #2855, total 4E-BP1 #9644, cleaved caspase-3 #9664, cleaved PARP #9544, pAKT$^{T473}$ #4060, TSC2 #4308, β-actin #4970; all from Cell Signaling Technology and used at 1:1000 dilutions. All uncropped western blots can be found in Supplementary Fig. 9.

**Molecular biology**. Total RNA was isolated using RNeasy (Qiagen) from three independent experiments. Labelling and hybridisation to the mouse Gene 1.0 ST Array system (Affymetrix), as well as data acquisition was performed by UCL Genomics at the Institute of Child Health. Normalisation and statistical analysis of the resulting array data was performed using GeneSpring software. Microarray data was also analysed by moderated *t*-test implemented by limma R package software[56] and a *p*-value cut-off of 0.05 was assigned for input into IPA.

Total cell mRNA was extracted using RNeasy (Qiagen). cDNA was synthesised using Superscript III reverse transcriptase (Invitrogen), and SYBR Green Master Mix (Qiagen) was used for qPCR reaction. The following primers were used: *Nanog* F, CTTACAAGGGTCTGCTACTGAGATGC; *Nanog* R, TGCTTCCTGGCAAGG ACCTT; *Esrrb* F, AACCGAATGTCGTCCGAAGAC; *Esrrb* R, GTGGCTGAGGG CATCAATG; *Rex1* F, CGAGTGGCAGTTTCTTCTTGG; *Rex1* R, GACTCACTT CCAGGGGGCAC; *Klf4* F, ACCTATACCAAGAGTTCTCATC; *Klf4* R, TCTGGC ACTGAAAGGGCCGG; *p21* F CCTGTCACTGTCTTGTACCCT; *p21* R, GCGTT TGGAGTGGTAGAAATCT; *p53* F CACGTACTCTCCTCCCCTCAAT; *p53* R AACTGCACAGGGCACGTCTT; *beta-actin* F, CTAAGGCCAACCGTGAAAAG; and *beta-actin* R, ACCAGAGGCATACAGGGACA.

To generate *p53* null CRISPR lines, a *p53* guide (5′-GCAGACTTTTCGCCACA GCG was cloned into lentiCRISPRv2 plasmid. Viruses were generated by transfecting this plasmid along with helper plasmids VSV-G and psPAX2 into HEK293T packaging cells. After 48 h, the media from these cells was applied to ESCs with 4ug/ ml polybrene. Two rounds of infection, for 4 h and then overnight, were carried out. The cells were then selected using 2 μg/ml puromycin and plated at single-cell confluency. Clones were screened for loss of p53 protein by western blot.

To generate *Tsc2* null CRISPR lines, we designed guide RNAs targeting its GAP domain, disruption of which has been previously reported to result in loss of protein[57]. To generate tetraploid mutant lines, we first infected cells with lentiCas9-Blast, as before, and selected a Cas9-expressing clone using Blasticidin treatment (at 2 μg/ml). This was infected with pU6-sgRNA-EF1alpha-hygro, in which we cloned the gRNA 5′GACAAGAAACGGCACCT. The lentiCas9-Blast was a gift from Feng Zhang (Addgene plasmid # 52962) and the pU6-sgRNA EF1Alpha-puro-T2A-BFP was a gift from Jonathan Weissman (Addgene plasmid #60955), and modified to hygro by Matias de Vas, Imperial College London. Clones selected with hygromycin (150 μg/ml) were screened on western blot for loss of Tsc2. To generate *Tsc2*$^{-/-}$ clones in wild-type cells, we co-transfected: hCas9; gRNAs (5′ GAAGGCCGGCCTACCTCATT and 5′GCATGGCTCTTACAGGTACA) using gRNA cloning vector (both a gift from George Church; Addgene number #41824); and a hygromycin marker (Clontech). The cells were selected using hygromycin.

Retroviral infections were performed as described above. All shRNAs (TSC2 #1, AGCTGTTACCTTGACGAAT; TSC2 #2, TGGACTACAAGTGCAACCT; PTEN, CAGCTAAAGGTGAAGATAT) were cloned into a MSCV-miRE-IRES-GFP-puro vector, which was a kind gift from Johannes Zuber (IMP, Vienna, Austria).

**Statistical methods**. Statistical analysis was performed using GraphPad Prism software. Statistical methods used are indicated in the relevant figure legends. No randomisation or blinding was used in experiments. Sample sizes were selected based on the observed effects and listed in the figure legends.

**Data availability**. The raw data for wild-type and *Bmpr1a* mutant cells in separate culture and co-culture after three days in N2B27 have been deposited in the GEO database under accession code GSE109494. The authors declare that all data supporting the findings of this study are available within the article and its supplementary information files or from the corresponding author upon reasonable request.

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

## Acknowledgements

We would like to thank Joerg Burgstaller, Katie Lawlor, Ana Lima, Salvador Perez-Montero, Juan-Miguel Sanchez and members of the Cell Proliferation Group for critical discussion. We thank Stephen Rothery for guidance and advice with confocal microscopy and Marta Abreu Paiva for advice on immunofluorescence protocols. Gratitude also goes to Matias de Vas, Laki Buluwela, Barbara Pernaute for assistance with CRISPR, James Elliot and Robert Sampson, for performing cell sorts. Sarah Bowling is a recipient of a MRC-DTA PhD studentship. Research in Tristan Rodriguez lab was supported by the MRC project grant (MR/N009371/1) and by the British Heart Foundation centre for research excellence. Research in Jesús Gil's laboratory was funded by a MRC-BHF Cardiovascular Stem Cell Research Strategic Development Grant (G0901467) and core support from the Medical Research Council (MC_U120085810). Michael Schneider is supported by the British Heart Foundation Simon Marks Chair of Regenerative Cardiology (CH/08/002/29257) and BHF Centre for Research Excellence (RE/13/4/30184).

## Author contributions

S.B. carried out the majority of the experiments and contributed to experimental design. A.D.G., M.S., S.P., M.A. and M.S. performed experiments. J.P.M.B and M.D.S. provided intellectual and strategic input. T.R. and J.G. designed experiments and overviewed the project. S.B., T.R. and J.G. wrote the manuscript.

## Additional information

**Competing interests:** The authors declare no competing interests.

