## [Peer Review File · Nature Communications]

Reviewers' comments:

Reviewer #1 (Remarks to the Author):

In this manuscript, Bowling et al. reveal the novel molecular machinery whereby defective cell elimination is conducted. In the previous paper (Sancho et al., *Dev. Cell*, 2013), the authors' group demonstrated that BMPR1-deficient or tetraploid cells undergo apoptosis when confronted with wild-type cells at E6.5 mouse embryo. Following this study, in this paper they have gained an insight into the molecular mechanism governing the phenomenon and presented data suggesting that p53-mediated downregulation of mTOR activity in the loser cells is indispensable for the process of cell competition. Furthermore, they demonstrate that the p53/mTOR axis could play an important role in the elimination of suboptimal cells in mouse embryos; caspase inhibition results in the massive accumulation of aberrant cells with low mTOR activity, highlighting the importance of this molecular pathway to ensure tissue homeostasis in early embryonic development. Overall, experiments are carefully designed and performed, and the presented data provide new insights into the molecular mechanisms underlying cell competition during development. Therefore, I feel that it is suitable for publication in *Nat. Commun.*, if the following concerns are properly addressed.

Major points

1. The expression level of p53 in the loser cells is comparable between separate and co-culture conditions (Fig. 4A-C). These results apparently do not make sense and contradict the authors' conclusion that p53 lies upstream of mTOR and mediates the defective cell elimination. It is highly likely that there are other key molecule(s), probably in concert with p53, that modulate mTOR signaling during the process of cell competition.
2. The authors contend that epistatic relationship between mTOR and p53 is substantiated in mouse embryos. However, the present set of data is not sufficient enough to draw the conclusion. In the current format, the authors just examined the effect of p53 activation on mTOR activity in Fig. 6G, H. The authors need to present more *in vivo* data. For example, is the expression or activity of p53 elevated in pS6 low cells? Does p53 inhibition boost mTOR signal, resulting in the reduction of the rate of apoptosis? The authors should address at least one of these questions.

Minor points

1. There is apparent discrepancy between Supplementary Figure 2A and B. In Fig. S2B, pS6 level in separate culture is slightly decreased or no change in shTsc2#1 or shTsc2#2, compared with that in EV, which contradicts the data shown in Fig. S2A.

2. There are several typos or mislabeling (the examples are shown below). The authors should check the text and figures more carefully and thoroughly.

- page 4, line 92 'early mouse cell competition'?

- page 6, line 190 Figure 4A, should be Figure 4D?

- Supplementary Figure 2D, the data do not make sense. There should be mislabeling (separate/co-culture and siC/SiTsc2 should be exchanged?).

Reviewer #2 (Remarks to the Author):

In the manuscript "A p53/mTOR signaling axis drives fitness selection through cell competition during early mouse embryonic development", Bowling and colleagues demonstrate that differential levels of mTOR signaling mark relative fitness between cells upon exit from pluripotency. Using both embryonic stem cells (ESCs) and the mouse embryo, they show that mTOR acts downstream of p53 to signal fitness and mediate cell competition, resulting in the elimination of "less fit" cells through apoptosis. Furthermore, their data allow for estimation that 30% of epiblast cells are eliminated by cell competition during normal early post-implantation development.

Cell competition is likely an important mechanism that ensures integrity of development, and this study demonstrates the significance of mTOR signaling in this process. The manuscript is well-written and would be suitable for publication upon addressing a few issues described below.

Major issues:

- While differential fitness is well demonstrated by the authors in vitro, it is not as clear whether a similar mechanism underlies cell competition in vivo. The authors should investigate whether levels of mTOR and p53 signaling are indeed heterogeneous within the epiblast. Specifically, in addition to their quantification of total pS6 in the epiblast (Figure 6H), they should measure whether there is a correlation between levels of p53 and mTOR signaling activity, and cleaved caspase/TUNEL signal at the level of individual cells.

- According to the authors' model, if they inject 2n Tsc2^{-/-} cells into WT blastocysts, those cells would outcompete WT embryonic cells and induce apoptosis. This would provide in vivo

evidence for the authors' model. This would also allow for decoupling the effects of change in mTOR signaling and of 4n in Figure 5AB.

Minor issues:

- Overall more quantitative measurement of the represented results would be necessary to substantiate their interpretation. In particular the authors could measure the signal intensity in images and interpret results from replicated experiments. For instance, the intensity of pS6 signal in Figure 5A, the degree of chimera contribution and mTOR activity in Figure 5B. How is "low or no" pS6 defined in Figure 6D and F? What percentage of pS6-positive cells was quantified on Fig. 6C? I would suggest to present mTOR activity as a proportion of the total S6 rather than pS6 itself.
- Figure 1D and Supplementary Figure 1H: pS6 levels appear to be higher in the 'less fit' *Bmpr1a*^{-/-} cells and 4n cells relative to WT in separate culture conditions. Are these differences significant?
- Figure 1D should mention the day of co-culture at which sorting was performed.
- Figure 2A schematic diagram could be shown, if necessary, in Figure 1A.
- Figure 5AB: could the author examine the mTOR activity in 4n ES cells before injection, so that the reduction of mTOR activity is not due to the polyploidy itself?
- Figure 5E "Control" should be written as "WT" to make it clear.
- Figure 6: co-immunostaining for lineage markers (epiblast and/or visceral endoderm) as well as higher magnification in insets would be helpful to show that signals of interest (e.g. pS6) are indeed specific to epiblast cells.
- Scale bars are missing, e.g. Figure 1B and F, Figure 4A, Figure 6A and G.
- Figure numbers should be corrected in several places in the Results section: p.6 line 190 Figure 4A should be "4D", p.7 line 221 Figure 5E should be "5D-E", p.8 line 236 Figure 6C-E should be "6C-D", p.8 line 239 Figure 6C-D should be "6E-F", and p.8 line 247 Figure 6F-G should be "6G-H".
- Typos: remove one "levels" (line 228); In cell "culture" (line 231).
- Relating to the major comment above, it would be worth discussing in greater detail whether and how cell competition may be critical for in vivo development. Is there anything known about the phenotype of *Tsc2*^{-/-} embryos? Since development of *p53*^{-/-} embryos is largely normal, is it possible that other pathways feed into mTOR signaling?

Takashi Hiiragi

Reviewer #3 (Remarks to the Author):

In this manuscript Rodriguez and colleagues identify through transcriptome profiling mTor downregulation as a key mediator of loser cell elimination in 2 separate cell competition models (BMP mutant and tetraploid cells). They show that readouts of the mTor pathway are specifically downregulated during competition, that downregulation is sufficient to kill cells autonomously and that concersely restoring mTor rescues cell competition. In addition blocking apoptosis causes the accumulation of cells with low pS6 levels (an mTor target) both in vitro and in early mouse embryos suggesting that this process is conserved in vitro and in vivo and that mTor downregulation is a physiological mechanism for the elimination of subfit cells in vivo. The authors further go on to show that p53 activity is upstream of mTor activation and is required for cell competition.

This is a very nice and well executed study and provides fundamental new advance in understanding the mechanisms of cell competition through the identification of a new p53-mTor axis as a key signal that defines the loser status. At the same time it provides further compelling data to show that cell competition is widespread in early development and contributes remarkably to determine the final cellular composition of the developing embryo. Altogether I think the findings are important and of general interest and I am supportive of publication in Nature Communications.

Below are some comments that I would like to see addressed in the revised manuscript.

Major comments

Table with gene lists and accession numbers for microarray data have not been provided. Equally tables with functionally enriched pathways identified gene ontology have not been provided.

In many of the rescue experiments rescue is monitored by cell number and not by inhibition of apoptosis, so it is formally possible that changes in cell divisions are able to rescue cell numbers. 3 key sets of experiments where it would be nice to show rescue of apoptosis (i.e. reduction in the number of apoptotic cells seen during competition) are the rescue of competition upon tsc KO (in both 4n and bmp competition) and the rescue of competition upon p53 KO. In all 3 cases there is a little increase in fold change already autonomously (see Figure 3A and 3C and Figure 4G) and that

could contribute or in fact account entirely for the rescue in cell number observed during competition. For these 3 experiments showing a rescue in apoptosis would rule out this simple explanation.

Minor comments:

The p53 targets shown in the table are not the most common. p21 which is a main target is not on list but then it is shown to be downregulated upon p53 inhibition in bmp cells (suppl figure 6A) . Is it actually upregulated in the first place in bmp^{-/-} cells vs wt cells?

Gamma H2AX labels only double strand break, it is formally possible that other types of DNA damage (single strand) are present and are upstream of p53 elevation. Adjust conclusions accordingly.

More information on the statistical methods and thresholds used to the microarray analysis should be provided.

More information on image analysis should be provided. For example how was pS6 fluorescence intensity measured? Fluorescence per field or per cell? And how were cells identified and segmented ? manually automatically and with what program? And was the fluorescence quantifies on single sections or stacks?

Figure 1H legend describes the data as FACS data, but Figure 1C describes similarly appearing data as fluorescence measurements. Is this a mistake?

Figure 1G: in the co-culture data it shows that WT are lower than 4n cells . Is this a mistake?

Why are 4n tsc^{-/-} cells winners against wt cells whereas bmp^{-/-} tsc^{-/-} double mutant are not? Could the authors comment on why even though they use the same pathways hyperactivation of mtor induces super-competition in the 4n but not on the BMP competition models?

P53 activity is observed prior to competition and is not further increased during competition, yet mtor activity is specifically inhibited during competition (in a p53 dependent manner). This indicates

that p53 elevation is necessary but not sufficient to inhibit mTor. Could the authors discuss this point and speculate on what they think is further happening as cell compete to decrease mtor?

Line 190 : do the authors mean Figure 4D?

We thank the reviewers for their comments that have helped strengthen our manuscript. Below is a detailed response to the points that they have raised.

Reviewer #1 (Remarks to the Author):

Major points

1. The expression level of p53 in the loser cells is comparable between separate and co-culture conditions (Fig. 4A-C). These results apparently do not make sense and contradict the authors' conclusion that p53 lies upstream of mTOR and mediates the defective cell elimination. It is highly likely that there are other key molecule(s), probably in concert with p53, that modulate mTOR signaling during the process of cell competition.

This is a very good point and we have clarified the manuscript accordingly. Given our data we envisage two possible scenarios to explain the relationship between p53 and mTOR. First, it is possible that p53 elevation labels defective cells as losers, but when these cells are in a competitive environment a second unidentified factor becomes activated in these defective cells, and this mediates repression of mTOR and their elimination. Alternatively it is possible that during competition, p53 changes in activity to repress mTOR and induce defective cell elimination. Distinguishing between these possibilities will help elucidate if the interaction between p53 and mTOR is direct or not. These possibilities have now been clarified in the discussion (page 10, first paragraph of the text).

2. The authors contend that epistatic relationship between mTOR and p53 is substantiated in mouse embryos. However, the present set of data is not sufficient enough to draw the conclusion. In the current format, the authors just examined the effect of p53 activation on mTOR activity in Fig. 6G, H. The authors need to present more *in vivo* data. For example, is the expression or activity of p53 elevated in pS6 low cells? Does p53 inhibition boost mTOR signal, resulting in the reduction of the rate of apoptosis? The authors should address at least one of these questions.

We have now analysed the expression of p53 at 5.5dpc and found it to be heterogeneous (Figure 6H). However, when we tested if there was a correlation between the relative p53 and pS6 levels at the single cell level at this stage we failed to find any significance in the unperturbed embryo (Supplementary Figure 8D). One likely explanation for this is that p53 has a very short half-life and its expression oscillates, with apoptosis being induced only when protein levels are stabilized above a certain threshold (Batchelor et al., *Molecular Systems Biology* 2011 and Purvis et al, *Science* 2012). Furthermore, how quickly p53 levels increase in the cell also determines if apoptosis is induced or not (Paek et al., *Cell* 2016). For these reasons, to obtain a rapid stabilization of p53 and analyse how this affects mTOR activity we have used the Mdm2 inhibitor Nutlin. When this was done we observed a direct correlation at the single cell level between p53 and pS6 levels (Figure 6I-J). This suggests that in the embryo, sustained p53 expression is required to repress mTOR. We have now discussed this in page 8 of the manuscript, 2nd paragraph.

Unfortunately we have found that Pifithrin- α , a common inhibitor used to block p53, does not decrease p53 target gene expression in ESCs (data not shown). For this reason we have not tried this inhibitor in embryos. Given that post-implantation mouse epiblast cells are remarkably difficult to transfect, we have also been unable to knock-down p53 function *in vivo* using siRNAs. We feel that the genetic experiment is out of the scope of what we can do for a revision, given the time it takes to import mice and perform the breeding experiments.

Additionally p63 and p73 appear to be able to compensate for p53 during early mouse embryogenesis (Wang et al., Cell Stem Cell 2017), further complicating the genetic inhibition experiment. For these reasons we have not been able to test if p53 inhibition boosts mTOR in the embryo.

Minor points

1. There is apparent discrepancy between Supplementary Figure 2A and B. In Fig. S2B, pS6 level in separate culture is slightly decreased or no change in shTsc2#1 or shTsc2#2, compared with that in EV, which contradicts the data shown in Fig. S2A.

In the original manuscript, the western blot lysates were from cells in pluripotent conditions whereas the flow cytometry plots were from differentiated cells (day 3 N2B27). We have now replaced the Western blot with lysates which were extracted from cells cultured in N2B27 cells and the data now match the flow cytometry plots.

2. There are several typos or mislabeling (the examples are shown below). The authors should check the text and figures more carefully and thoroughly.

- page 4, line 92 'early mouse cell competition'?

- page 6, line 190 Figure 4A, should be Figure 4D?

- Supplementary Figure 2D, the data do not make sense. There should be mislabeling (separate/co-culture and siC/SiTsc2 should be exchanged?).

We apologise for these errors that have now been corrected.

Reviewer #2 (Remarks to the Author):

Major issues:

1. While differential fitness is well demonstrated by the authors in vitro, it is not as clear whether a similar mechanism underlies cell competition in vivo. The authors should investigate whether levels of mTOR and p53 signaling are indeed heterogeneous within the epiblast. Specifically, in addition to their quantification of total pS6 in the epiblast (Figure 6H), they should measure whether there is a correlation between levels of p53 and mTOR signaling activity, and cleaved caspase/TUNEL signal at the level of individual cells.

We have now included data showing that p53 expression is heterogeneous in the epiblast at 5.5dpc (Figure 5H). However, although we have also found pS6 levels of expression to be heterogeneous in the unperturbed embryo (Supplementary Figure 8), we found no correlation between the expression of p53 and pS6 (Supplementary Figure 8D). One likely explanation for this is that p53 has a very short half-life and its expression oscillates, with apoptosis being induced only when protein levels are stabilized above a certain threshold (Batchelor et al., Molecular Systems Biology 2011 and Purvis et al, Science 2012). Furthermore, how quickly p53 levels increase in the cell also determine if apoptosis is induced or not (Paek et al., Cell 2016). For these reasons, to obtain a rapid stabilization of p53 and analyse how this affects mTOR activity we have used the Mdm2 inhibitor Nutlin3a. When this was done we observed a direct correlation at the single cell level between p53 and pS6 levels (Figure 6G-H). This suggests that in the embryo, sustained p53 expression is required to repress mTOR. We have now discussed this in page 8, 2nd paragraph of the manuscript.

[redacted]

2. According to the authors' model, if they inject 2n Tsc2^{-/-} cells into WT blastocysts, those cells would outcompete WT embryonic cells and induce apoptosis. This would provide in vivo evidence for the authors' model. This would also allow for decoupling the effects of change in mTOR signaling and of 4n in Figure 5AB.

[redacted]

Minor issues:

1. Overall more quantitative measurement of the represented results would be necessary to substantiate their interpretation. In particular the authors could measure the signal intensity in images and interpret results from replicated experiments. For instance, the intensity of pS6 signal in Figure 5A, the degree of chimera contribution and mTOR activity in Figure 5B. How is "low or no" pS6 defined in Figure 6D and F? What percentage of pS6-positive cells

was quantified on Fig. 6C? I would suggest to present mTOR activity as a proportion of the total S6 rather than pS6 itself.

We have now included the quantifications requested.

We have quantified in Supplementary Figure 7A-B the intensity of pS6 signal of the embryos shown in Figure 5A and B.

We have quantified the level of 4n and 4n-Tsc2^{-/-} contribution and this is now shown in supplementary Figure 7C.

To measure pS6 in Figure 6D and F we draw a line through the cytoplasm of each epiblast cell at the basal side and the maximum fluorescence (grey value) across that line was measured. To define low pS6, a cut-off of 20 000 grey values (litter 1; n=9), 12 000 grey values (litter 2; n=9) or 10 000 grey values (litter 3; n=10) was used to define low pS6. We adjusted the cut off between litters owing to variability in mean fluorescence levels between independent experiments. A ratio cut off of 0.3 was used to define the low pS6:S6 population. Example graphs showing the cut-off used are now presented in Supplementary Figure 8A-B and this now explained in the materials and methods section.

The relative levels of pS6 to the total S6 levels are now presented in Figure 6F and Supplementary Figure 8B-C.

2. Figure 1D and Supplementary Figure 1H: pS6 levels appear to be higher in the ‘less fit’ Bmpr1a^{-/-} cells and 4n cells relative to WT in separate culture conditions. Are these differences significant?

These differences are significant and this now indicated in figure 1E.

3. Figure 1D should mention the day of co-culture at which sorting was performed.

This is now indicated in the Figure.

4. Figure 2A schematic diagram could be shown, if necessary, in Figure 1A.

A schematic has now been included.

5. Figure 5AB: could the author examine the mTOR activity in 4n ES cells before injection, so that the reduction of mTOR activity is not due to the polyploidy itself?

Figure 1F-H shows pS6 levels in 4n cells. In separate culture 4n cells have higher pS6 levels than controls, as shown by the flow cytometry analysis, indicating that ploidy does not affect S6 phosphorylation.

6. Figure 5E “Control” should be written as “WT” to make it clear.

This has now been changed to WT.

7. Figure 6: co-immunostaining for lineage markers (epiblast and/or visceral endoderm) as well as higher magnification in insets would be helpful to show that signals of interest (e.g. pS6) are indeed specific to epiblast cells.

We have now shown a higher magnification of the images provided, showing that the pS6 staining is higher in the inner epiblast cells (Figure 1E).

8. Scale bars are missing, e.g. Figure 1B and F, Figure 4A, Figure 6A and G.
We have added scale bars.

9. Figure numbers should be corrected in several places in the Results section: p.6 line 190 Figure 4A should be “4D”, p.7 line 221 Figure 5E should be “5D-E”, p.8 line 236 Figure 6C-E should be “6C-D”, p.8 line 239 Figure 6C-D should be “6E-F”, and p.8 line 247 Figure 6F-G should be “6G-H”.

Where appropriate these have been corrected.

10. Typos: remove one “levels” (line 228); In cell “culture” (line 231).

This has been corrected.

11. Relating to the major comment above, it would be worth discussing in greater detail whether and how cell competition may be critical for in vivo development. Is there anything known about the phenotype of Tsc2^{-/-} embryos? Since development of p53^{-/-} embryos is largely normal, is it possible that other pathways feed into mTOR signaling?

We have discussed possible roles of cell competition during post-implantation development in pages 10 and 11 of the manuscript. We have also included what is known about the Tsc2 null phenotype and discussed the possibility that like occurs during mesendoderm induction (Wang et al., Cell Stem Cell 2017), p63 and p73 may compensate for p53 in the early mouse embryo (page 11, lines 364-372).

Reviewer #3 (Remarks to the Author):

Major comments

Table with gene lists and accession numbers for microarray data have not been provided. Equally tables with functionally enriched pathways identified gene ontology have not been provided.

We have now been provided tables with those factors with enriched targets found from the microarray analysis of defective cells (Supplementary Figure 5A). The accession number is also provided in the material and methods section (GSE109494). We were unsure if the reviewer wanted additional information in the table, and if so we would be happy to add this information if requested.

In many of the rescue experiments rescue is monitored by cell number and not by inhibition of apoptosis, so it is formally possible that changes in cell divisions are able to rescue cell numbers. 3 key sets of experiments where it would be nice to show rescue of apoptosis (i.e. reduction in the number of apoptotic cells seen during competition) are the rescue of competition upon tsc KO (in both 4n and bmp competition) and the rescue of competition upon p53 KO. In all 3 cases there is a little increase in fold change already autonomously (see Figure 3A and 3C and Figure 4G) and that could contribute or in fact account entirely for the rescue in cell number observed during competition. For these 3 experiments showing a rescue in apoptosis would rule out this simple explanation.

We now include data showing that inhibition of Tsc2 rescues the apoptosis increase observed in 4n and Bmpr1a^{-/-} ESCs during cell competition (Figure 3A and C and Figure 4H).

Minor comments:

The p53 targets shown in the table are not the most common. p21 which is a main target is not on list but then it is shown to be downregulated upon p53 inhibition in bmp cells (suppl figure 6A) . Is it actually upregulated in the first place in bmp^{-/-} cells vs wt cells?

We have now included data showing that p21 mimics p53 expression, and is elevated in Bmpr1a^{-/-} cells in separate and co-culture conditions (Supplementary Figure 5B).

Gamma H2AX labels only double strand break, it is formally possible that other types of DNA damage (single strand) are present and are upstream of p53 elevation. Adjust conclusions accordingly.

We have clarified that we refer to double strand DNA breaks (page 6, line 188-190).

More information on the statistical methods and thresholds used to the microarray analysis should be provided.

We have now provided this information in the materials and methods section (page 14, line 483-484).

More information on image analysis should be provided. For example how was pS6 fluorescence intensity measured? Fluorescence per field or per cell? And how were cells identified and segmented ? manually automatically and with what program? And was the fluorescence quantifies on single sections or stacks?

In cells, levels of cytoplasmic pS6 per field were quantified by measuring average grey value in the two pixels surrounding cell nuclei, using DAPI as a mask. Levels of nuclear p53 per field were quantified by measuring average grey value in cell nuclei, using DAPI as a mask. This was performed in ImageJ on single sections. In both cases, a minimum of three fields were used for each experiment.

For pS6 and p53 quantification in embryos, a line was drawn manually across the nucleus and cytoplasm of each cell and the maximum grey value across each line was measured. For cleaved caspase-3 quantification of tetraploid cells in embryos, cells positive or negative for cleaved caspase-3 were counted manually. In all cases, the analysis was performed in ImageJ on single sections.

We have now provided this information in the materials and methods section (page 13).

Figure 1H legend describes the data as FACS data, but Figure 1C describes similarly appearing data as fluorescence measurements. Is this a mistake?

This has now been clarified in the figure legend.

Figure 1G: in the co-culture data it shows that WT are lower than 4n cells . Is this a mistake?

We have clarified the labels to highlight that pS6 levels are lower in 4n cells in co-culture.

Why are 4n tsc^{-/-} cells winners against wt cells whereas bmp^{-/-} tsc^{-/-} double mutant are not? Could the authors comment on why even though they use the same pathways hyperactivation of mtor induces super-competition in the 4n but not on the BMP competition models?

We are sorry that the way we organised the data in the figures created confusion. What we observe is that both in Bmpr1a^{-/-} and 4n cells, CRISPR mutation of Tsc2 makes them into

super-competitors. In contrast to this shRNA or siRNA inhibition rescues defective cell elimination during competition but does not make these cells into super-competitors. This is now discussed in page 5 of the text. We have also moved the CRISPR data to the main figures and the sh/siRNA inhibition to the supplementary figures.

P53 activity is observed prior to competition and is not further increased during competition, yet mtor activity is specifically inhibited during competition (in a p53 dependent manner). This indicates that p53 elevation is necessary but not sufficient to inhibit mTor. Could the authors discuss this point and speculate on what they think is further happening as cell compete to decrease mtor?

This is a very good point and was also raised by reviewer 1. We have now discussed this point and speculated on what changes could occur during competition to decrease mTOR (page 10, first paragraph of the text).

Line 190 : do the authors mean Figure 4D?

This has now been corrected.

REVIEWERS' COMMENTS:

Reviewer #1 (Remarks to the Author):

In the revised manuscript, the authors have fully addressed my concerns. Thus, I think that the paper is now acceptable for Nature Communications.

--

Reviewer #2 (Remarks to the Author):

The authors addressed the issues raised by this reviewer. This study is suitable for publication in Nature Communications.

--

Reviewer #3 (Remarks to the Author):

I find the revised manuscript overall much improved in clarity and strengthened with additional data. My criticisms have all been addressed satisfactorily.

I have no further concerns and recommend that the manuscript be accepted publication.